# Neutral equilibrium and forcing feedbacks in marine ice sheet modelling

Rupert Gladstone[1], Yuwei Xia[2], and John Moore[1,2]

[1]Arctic Centre, University of Lapland, Rovaniemi, 96101, Finland
[2]College of Global Change and Earth System Science, Beijing Normal University, 100082, China

*Correspondence to:* Rupert Gladstone (rupertgladstone1972@gmail.com)

**Abstract.**

Poor convergence with resolution of ice sheet models when simulating grounding line migration has been known about for over a decade. However, some of the associated numerical artefacts remain absent from the published literature.

In the current study we apply a Stokes-flow finite element marine ice sheet model to idealised grounding line evolution experiments. We show that with insufficiently fine model resolution, a region containing multiple steady state grounding line positions exists, with one steady state per node of the model mesh. This has important implications for the design of perturbation experiments used to test convergence of grounding line behaviour with resolution. Specifically, the design of perturbation experiments can be under-constrained, potentially leading to a "false positive" result. In this context a false positive is an experiment that appears to achieve convergence when in fact the model configuration is not close to its converged state. We demonstrate a false positive: an apparently successful perturbation experiment (i.e. reversibility is shown) for a model configuration that is not close to a converged solution. If perturbation experiments are to be used in the future, experiment design should be modified to provide additional constraints to the initialisation/spin-up requirements.

This region of multiple locally stable steady state grounding line positions has previously been mistakenly described as neutral equilibrium. This distinction has important implications for understanding the impacts of discretizing a forcing feedback involving grounding line position and basal friction. This forcing feedback can not, in general, exist in a region of neutral equilibrium, and could be the main cause of poor convergence in grounding line modelling.

## 1 Introduction

Strongly resolution dependent behaviour when implementing grounding line movement (sometimes referred to as grounding line migration) in a marine ice sheet model was identified by Vieli and Payne (2005) and was further characterised as a convergence problem by subsequent studies (Durand et al., 2009; Goldberg et al., 2009; Gladstone et al., 2010a, b, 2012). In the current study "convergence" refers to the approach of model outputs to a consistent state as resolution is made progressively finer. Some models incorporating a moving grid that explicitly tracks grounding line position do not appear to exhibit this poor convergence (Vieli and Payne, 2005). Various forms of mesh refinement help to address the problem, though very high resolution is still needed (Goldberg et al., 2009; Cornford et al., 2013), and special treatments of the grid cell or element

containing the grounding line can also improve convergence (Pollard and DeConto, 2009; Gladstone et al., 2010b; Gagliardini et al., 2016; Seroussi et al., 2014; Feldmann et al., 2014).

This problem has also been decribed as neutral equilibrium (Durand et al., 2009; Pattyn et al., 2006) in modelling studies. This terminology may follow from earlier studies in which it was proposed that real marine ice sheet systems may exhibit neutral equilibrium (Hindmarsh, 2006). Although these theories are no longer accepted (Schoof, 2007), unconverged model behaviour at coarse resolution is still sometimes referred to as neutral equilibrium (Durand et al., 2009).

Most of the studies cited above use a Weertman sliding relation (Weertman, 1957). More recent studies (Leguy et al., 2014; Tsai et al., 2015; Gladstone et al., 2017) suggest that the convergence issues may be to some extent mitigated by use of sliding relations incorporating a dependence on effective pressure at the bed. However, irrespective of sliding law, similar convergence issues may arise due to a step change in basal melting at the grounding line (Gladstone et al., 2017). The numerical implementation of basal melting at the element or grid cell scale may also have a large impact on convergence (Seroussi and Morlighem, 2018).

In the current study we employ a flowline Stokes-flow model with Weertman sliding and no basal melting (Section 2) to further characterise the nature of this grounding line convergence issue (Section 3). We choose a setup in which the problem is under-resolved, i.e. the model outputs are not close to a converged solution. This is chosen in order to demonstrate the nature of the numerical artefacts arising at coarse resolution. We explore implications for design of computer experiments (Section 3.1) and for the issue of neutral equilibrium (Section 4). This leads to a discussion on discretisation of a forcing feedback involving basal friction and model state (Section 5).

## 2 Flowline modelling

Our aim is to provide a model configuration in which convergence with resolution is not achieved (i.e. our resolution is too coarse for self-consistent model behaviour), and explore the nature of the grounding line problems. Our setup is similar to that of the original Marine Ice Sheet Model Intercomparison Project (Pattyn et al., 2012), with a linear bed, Weertman sliding, and spatially uniform surface accumulation.

We use the ice dynamic model (IDM) Elmer/Ice (Gagliardini et al., 2013). The Stokes equations for a viscous fluid with non-linear rheology are solved using the finite element method over a two-dimensional flowline domain (one vertical and one horizontal dimension) with grounding line capability.

A contact problem is solved to determine the evolving grounding line position (Favier et al., 2012), which is constrained to be located at a node. Basal resistance, or friction, in the vicinity of the grounding line is determined using the discontinuous approach (DI in Gagliardini et al. (2016)). This imposes full friction for all grounded elements and free-slip for all floating elements, with the element containing both grounded and floating nodes considered to be floating.

The rheology follows Glen's law (Glen, 1952; Paterson, 1994) with viscosity calculated using a temperature dependent Arrhenius law (Gagliardini et al., 2013; Paterson, 1994). A constant uniform temperature of -15 C is used in all simulations.

**Table 1.** Summary of experiments.

| Experiment | Initial condition | Advance phase | Retreat phase |
| --- | --- | --- | --- |
| ARXX | Uniform 100m slab | Run length= $7\,\text{ka}$, $a =\text{XX}\,\text{ma}^{-1}$ | Run length= $6\,\text{ka}$, $a = 0.2\,\text{ma}^{-1}$ |
| P1 | AR0.7 final state | Run length= $1\,\text{ka}$, $a = 2.0\,\text{ma}^{-1}$ | Run length= $4\,\text{ka}$, $a = 0.2\,\text{ma}^{-1}$ |
| P2 | AR1.7 final state | Run length= $1\,\text{ka}$, $a = 2.0\,\text{ma}^{-1}$ | Run length= $4\,\text{ka}$, $a = 0.2\,\text{ma}^{-1}$ |
| SP | AR0.7 final state | Run length= $1\,\text{ka}$, $a = 0.71\,\text{ma}^{-1}$ | Run length= $4\,\text{ka}$, $a = 0.7\,\text{ma}^{-1}$ |

The linear down sloping bedrock, $b$, is given in $\text{m}$ relative to sea level by

$$b = 500 - 0.005 \times x, \tag{1}$$

where $x$ is distance from the inland boundary. The domain length is $600\,\text{km}$.

The horizontal component of the velocity is set to zero at the inland boundary, and an ocean pressure condition applied at
the ice front and under the floating ice shelf. We take ice density to be $910\,\text{kgm}^{-3}$ and water density to be $1000\,\text{kgm}^{-3}$.

A spatially uniform net surface accumulation flux, $a$, is used, and this value is varied between simulations.

The basal friction or shear stress, $\tau_b$, acts opposite to the direction of flow and has magnitude (Weertman, 1957)

$$\tau_b = C u_b^{\frac{1}{3}} \tag{2}$$

where $u_b$ is the sliding velocity and $C$ is a friction coefficient. $C$ is set to $0.02417\,\text{MPam}^{-\frac{1}{3}}\text{a}^{\frac{1}{3}}$ for all simulations in the current
study.

The simulations carried out for the current study are described below. They comprise grounding line advance simulations
followed by grounding line retreat simulations (Section 2.1). In some cases further perturbation experiments are then carried
out (Section 2.2). These simulations were all carried out with a horizontally uniform element size of $1\,\text{km}$ and a timestep size
of $0.2\,\text{a}$. Typical steady state profiles for this model setup are shown in Figure 1.

**2.1   Advance/retreat experiments**

The advance simulations are spun-up from a uniform slab of 100m thickness. They comprise $7\,\text{ka}$ of evolution with a different
net accumulation forcing for each simulation. Values range from $0.2\,\text{ma}^{-1}$ to $2.0\,\text{ma}^{-1}$ (the full set of values used is given in
the Figure 2 legend).

The advance simulations are followed immediately by "retreat" simulations, which in some (but not all) cases exhibit ground-
ing line retreat. These simulations continue from the final states of the advance simulations. They all use a net accumulation
forcing of $a = 0.2\,\text{ma}^{-1}$ and are run for a further $6\,\text{ka}$.

Each pair of advance and retreat simulations constitutes an ARXX experiment, where XX indicates the accumulation forcing
used for the advance phase (e.g. AR0.7 refers to the advance simulation with $a = 0.7\,\text{ma}^{-1}$ and the corresponding retreat
phase). Experiments are summarised in Table 1.

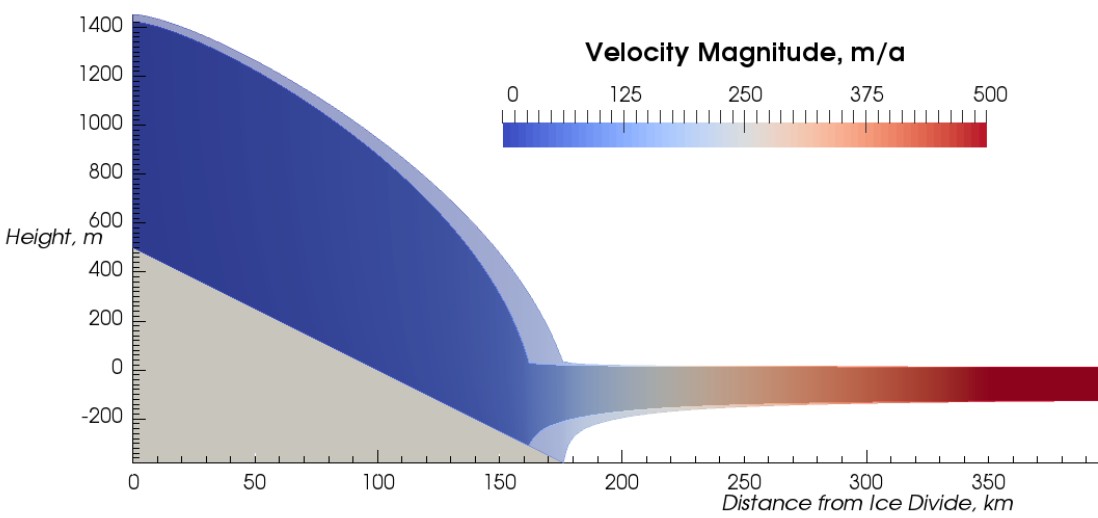

**Figure 1.** Ice sheet profiles after $13\,\mathrm{ka}$ at the end of the advance/retreat simulations described in Section 2.1. The simulations with an advance forcing of $a = 0.7\,\mathrm{ma}^{-1}$ (solid colour) and $a = 1.7\,\mathrm{ma}^{-1}$ (semi-transparent) are shown. These states provide the initial state for perturbations experiments P1 and P2 respectively (Section 2.2). Vertical exaggeration is 100 times.

## 2.2 Perturbation experiments

Starting from the final states of two of the advance/retreat simulations, (i.e. after $13\,\mathrm{ka}$ total simulation time) we carried out two perturbation simulations. P1 starts from the final state of experiment AR0.7 (which used an advance forcing of $a = 0.7\,\mathrm{ma}^{-1}$), and P2 starts from the final state of AR1.7. AR0.7 and AR1.7 are shown with a dotted line in Figure 2 and their final states (which form the initial states for P1 and P2) are shown in Figure 1. Note that although P1 and P2 start from approximate steady states, and in both cases the steady state was approached with $a = 0.2\,\mathrm{ma}^{-1}$, these steady states are distinct. The forcing for the perturbation experiments P1 and P2 is identical apart from initial state. They are run for $1\,\mathrm{ka}$ with $a = 2.0\,\mathrm{ma}^{-1}$ followed by $4\,\mathrm{ka}$ with $a = 0.2\,\mathrm{ma}^{-1}$. The perturbation experiments P1 and P2 are shown in Figure 3.

We also carried out a small perturbation experiment, PS. PS starts from the same state as P1. It is identical to P1 except that the magnitude of the perturbed forcing is $a = 0.71\,\mathrm{ma}^{-1}$ and the retreat phase has $a = 0.7\,\mathrm{ma}^{-1}$. Experiments are summarised in Table 1.

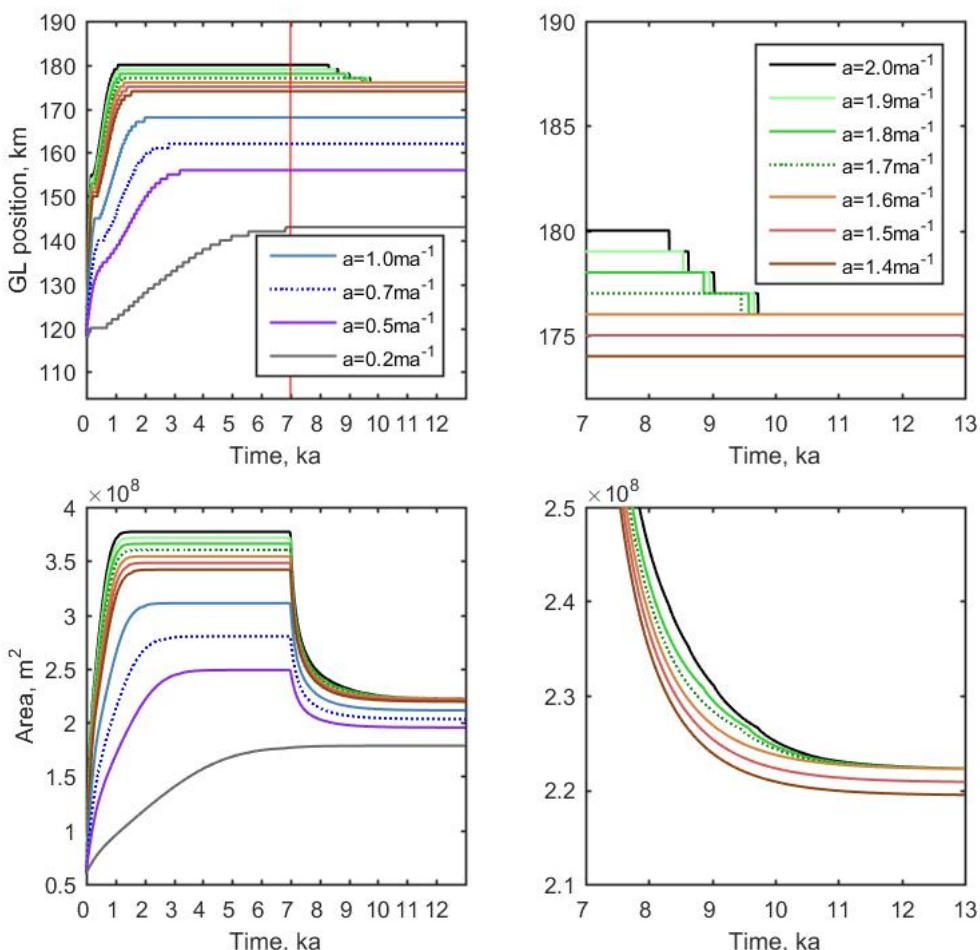

**Figure 2.** Advance/retreat simulations (described in Section 2.1). Evolution of (top panel) grounding line position for all simulations and (lower panel) ice area. Area in this case is the flowline equivalent of ice volume for a 3D ice sheet, and can also be interpreted as volume per unit width of the glacier. The right hand subplots show detail of a subset of the retreat simulations. The red vertical line indicates the forcing change at $7\,\mathrm{ka}$ when the simulations switch from advance to retreat. The legend shows the accumulation rates prescribed during the advance phase, while for the retreat accumulation was $0.2\,\mathrm{ma}^{-1}$. The dotted lines are the cases also shown in Figure 1 and provide the initial state for the P1 and P2 experiments (described in Section 2.2).

## 3 Multiple steady states

Figure 2 summarises the evolution over time of the advance/retreat simulations. Although a formal test for steady state was not imposed, we calculated the rate of change of area and found it to be of the order of $10^{-8}\,\mathrm{m^2a^{-1}}$ or smaller at the end of all advance simulations and all retreat simulations (except for AR0.2 for which we calculated the rate only at the end of the retreat

simulation as it was not yet at steady state after 7 ka). Given that 7 ka years is sufficient to approach steady state, retreat should occur after 7 ka years for all simulations in which $a$ was initially greater than $0.2\,\mathrm{ma}^{-1}$. This is due to the uniqueness of stable ice sheet configurations on a linear down sloping bed, demonstrated for a "shelfy-stream" approximation by Schoof (2007). However, as also seen with a "shelfy-stream" model (Gladstone et al., 2010a), multiple steady states exist as a model artefact.

The multiple steady states that exist after 13 ka are almost certainly numerical artefacts, with the underlying system having only one viable steady state. Similar studies have shown that the size of this region decreases with finer resolution (Gladstone et al., 2010a). The region containing steady state grounding line positions (at the end of the retreat phase) in the current study spans from $x = 143\,\mathrm{km}$ to $x = 176\,\mathrm{km}$. We propose that the model is capable of exhibiting as many viable steady state grounding line positions as there are mesh nodes within this region. We tested this hypothesis near the seaward end of the

region by implementing small increments in $a$ between advance simulations (experiments AR1.4 up to AR2.0). Specifically we obtained a final grounding line position on every node from $x = 174\,\mathrm{km}$ to $x = 180\,\mathrm{km}$ for the advance simulations and from $x = 174\,\mathrm{km}$ to $x = 176\,\mathrm{km}$ for the retreat simulations (Figure 2, upper right panel).

      The volume evolution plots indicate a reduction in volume for all retreat simulations (except the simulation which advanced under $a = 0.2\,\mathrm{ma}^{-1}$ forcing), even for simulations showing no grounding line movement (experiments AR0.2 up to AR1.6).

Simulations ending the retreat phase with the same grounding line position (experiments AR1.6 up to AR2.0 end at $176\,\mathrm{km}$) have the same final volume. Simulations with a more landward final grounding line position have a lower final volume. Thus, for our setup, $176\,\mathrm{km}$ marks the seaward end of the region of steady state grounding line positions under a forcing of $a = 0.2\,\mathrm{ma}^{-1}$. Our explanation for this numerical artefact involves discretisation of a feedback between model state (especially grounding line position) and total basal resistance, which we discuss in Section 5.

## 3.1   Implications for experiment design

      Here we consider perturbation experiments P1 and P2, both of which adhere to a typical perturbation design and both of which experience identical forcing during the experiment. Perturbation experiments are common in IDM studies and intercomparison projects (e.g. Pattyn et al. (2006, 2012, 2013); Favier et al. (2012)). The premise is that an initial spin-up procedure results in an IDM in steady state. A forcing perturbation is applied, causing change, and then removed. The analysis then considers

whether or not reversibility has been demonstrated (i.e. whether the IDM state returned to its post spin-up state after the forcing perturbation was reset). However, the existence of multiple steady state grounding line positions means that the requirement to start in steady state is not sufficient to constrain the initial (post-spin-up) state.

      The outputs of the perturbation experiments are shown in Figure 3. Although both experiments adhere to typical perturbation experiment design, and are both subject to the same perturbation, P2 shows full reversibility (in all aspects of model state, including grounding line position, ice volume and total friction) and P1 does not. The outcomes in terms of reversibility are

opposites, resulting directly from the choice of initial state. Here, P2 is a false positive, because considered in isolation it would appear to indicate a converged result, but convergence has not been achieved. In general a converged experiment (i.e. with sufficiently fine resolution to achieve a self consistent result) will always demonstrate reversibility on a linear bed (Schoof, 2007), but our results demonstrate that reversibility is not in itself a sufficient criterion to establish convergence.

We now consider an example of this vulnerability in design of perturbation experiments from the published literature. Pattyn et al. (2006) investigated the role of transition zones in grounding line modelling. The transition zone is a region immediately upstream of the grounding line over which the stress state changes from a grounded regime (in which high basal shear stress approximately balances gravitational driving stress) to a floating regime (where basal shear stress is zero and longitudinal stress in the ice balances a low gravitational driving stress).

Pattyn et al. (2006) used a spin-up procedure that resulted, for most of their simulations, in retreat of the grounding line as steady state was approached. This suggests (but does not prove) that the end of the spin-up resulted in a steady state grounding line position located at the seaward end of the region of multiple steady states, analogous to our experiment P2 (Figure 3). These simulations did demonstrate reversibility. However, their simulation with a short prescribed grounding line transition zone involved no movement of the grounding line as steady state was approached. This suggests (but again does not prove) that the end of the spin-up resulted in a steady state grounding line position somewhere within the region of multiple steady states, analogous to our experiment P1 (Figure 3), see Figure 4 of Pattyn et al. (2006). This simulation did not demonstrate reversibility. Thus the result of Pattyn et al. (2006) that a longer transition zone results in better reversibility may be an artefact of their experiment design rather than a robust result.

### 3.1.1 Initialisation through inversion

A common method for initialisation of IDMs is to infer basal properties and ice viscosity (or temperature) through inverse techniques, steady state temperature simulations, and surface relaxation (Morlighem et al., 2010; Gillet-Chaulet et al., 2012; Gladstone et al., 2014; Cornford et al., 2015; Gillet-Chaulet et al., 2016; Zhao et al., 2018). These methods can lead to an initial state that is close to steady state, but without any information as to whether convergence is achieved, and hence whether multiple steady states may exist. If a transient simulation intialised in such a way leads to little or no change in terms of grounding line position, it cannot be concluded that the real system being modelled is close to steady state, because the possiblity remains that the modelled steady state is a result of under-resolution.

If a convergence study is carried out for the system undergoing retreat, for example if high basal melt forcing is applied (Favier et al., 2014), this does not prove that the model will also exhibit converged beahviour in advance, nor even does it prove that regions of stationary grounding line within the domain are indicative of a converged result. Conversely, convergence of advance behaviour does not prove convergence of retreat behaviour. The current study does not fully explore the implications of multiple steady grounding line positions in 3D real world cases, which is espeically complicated in that some regions may exhibit an advancing grounding line while others exhibit a retreating grounding line. Based on current simulations the authors recommend that both advance and retreat convergence tests be carried out in order to have full confidence in modelled behaviour.

## 4 Neutral equilibrium

An equilibrium state (or steady state) of a system is a state that does not change unless the forcing changes. In the context of IDM grounding line simulations, this means that neither the forcing applied to the domain nor the ice sheet configuration are changing over time. Such steady states are typically obtained through long simulations in which forcing is kept constant and the state of the simulated ice sheet gradually stops evolving as equilibrium is approached.

It is important to clarify different types of equilibria for the following discussion. Consider the example of a ball at rest (i.e. in equilibrium) under gravitation on a solid surface. The ball is then subjected to a perturbation: it is moved along the surface then left only under gravitation. Different types of equilibrium may be illustrated by considering the behaviour of the ball after the perturbation has been removed.

An equilibrium state where the perturbation results in the system tending to return to the original state is a stable equilibrium (e.g. Figure 4a - the ball rolls back down to the original position).

An equilibrium state where the perturbation results in the system tending to move further from the original state is an unstable equilibrium (not shown, but consider a ball perched on the summit of a hill - it will continue rolling away from the summit after being given a small push in any direction).

An equilibrium state where the perturbation results in the system remaining in the new state is termed neutral equilibrium (e.g. Figure 4b - the ball may be at rest anywhere on the flat region).

Figure 4c illustrates a system with multiple locally stable equilibria within a confined region. A sufficiently large perturbation will result in the ball finding a new equilibrium position, but a small perturbation will result in a return to the original position.

These types of behaviour for a ball under gravity have analogies for a marine ice sheet system. Schoof (2007) demonstrated the existence of a single stable equilibrium for a marine ice sheet on a downward (in the ice flow direction) sloping bed. Schoof (2007) also demonstrated the existence of an unstable equilibrium on an upward sloping bed, though this may not always be the case in the presence of high lateral drag (Katz and Worster, 2010; Gudmundsson et al., 2012). As mentioned in Section 1, neutral equilibrium in real world marine ice sheet systems is no longer considered plausible, but multiple stable equilibria could exist as a function of bedrock geometry (Schoof, 2007).

IDM studies using different models have demonstrated that multiple steady states can, as a numerical artefact, exist in models where the system being modelled should exhibit a single stable equilibrium (Durand et al., 2009; Gladstone et al., 2010a). This has been referred to as neutral equilibrium (Durand et al., 2009), and here we consider the distinction between a region of neutral equilibrium and a region of multiple locally stable steady states. We argue that IDMs exhibit, as a numerical artefact, a region containing multiple locally stable equilibria (similar to Figure 4c) and not a region of neutral equilibrium.

Figure 5 shows output from experiment PS, the small perturbation experiment. The perturbation, although not sufficient to cause a change in grounding line position (Figure 5a), is sufficient to cause a shift in model state, as evidenced by the change in total ice volume (Figure 5c). However, the forcing reset results in a return to the original state. This behaviour indicates a locally stable steady state rather than a region of neutral equilibrium. This argument against marine IDMs exhibiting neutral

equilibria may appear to be a matter of semantics, but there are important implications toward understanding the nature of the grounding line convergence problem, discussed in Section 5.

## 5 Forcing feedback

Figure 6 shows in more detail the evolution of total basal friction during the advance phase of perturbation experiments P1 and P2. The spikes in total friction correspond to advance of the grounding line by a single element. These features are characterised by an instananeous increase of total friction followed by a rapid decrease and an ensueing gradual increase. The spikes can be explained as follows: An instantaneous increase in basal friction results from a grounding line advance due to the increased contact area. This increase reduces the sliding velocity, causing the rapid decrease in total friction. This is followed by a more gradual return to the longer term trend.

This is a model discretization of what should be a continuous feedback: incremental grounding line advance should cause incremental increase in total basal friction, causing an incremental slowing and thickening. This positive feedback (which we refer to as the friction force feedback) between grounded extent and total friction is continuous in the underlying system being simulated, but heavily discretised in the model due to basal friction reaching a peak at the grounding line. The modelled flux across the grounding line must be higher than that of the system it attempts to represent in order to compensate for the missing basal friction immediately downstream of the grounding line due to the discretisation. Specifically, the PS experiment (Figure 5) demonstrates that even an increase in modelled volume and total friction force of several tens of percent may not be sufficient to cause a single element of grounding line advance. This understanding could not have been attained if the region of multiple locally stable steady states was viewed as a region of neutral equilibrium, because a neutral equilibrium can have no positive feedback between forcing and state (except in the vanishingly low probability case of an exactly compensating mechanism).

We postulate that this discretisation of a continuous feedback is the main cause of numerical artefacts and poor convergence with resolution in grounding line modelling. This is consistent with the finding that sliding relations incorporating a strong dependency on effective pressure at the bed show far better convergence with resolution (Gladstone et al., 2017). This is due to the basal friction approaching zero at the grounding line, so that the advance or retreat of the grounding line by a single element will not have a significant impact on total basal friction.

Considering several published sliding relations that feature a dependence on effective pressure, it should be noted that the hybrid sliding relations of Gagliardini et al. (2007) and Tsai et al. (2015) typically feature steep basal friction gradients over a transition zone near the grounding line (Brondex et al., 2017) and so may not exhibit such good convergence as the sliding relation of Budd et al. (1979, 1984). It should also be noted that improved convergence is not a valid reason to choose one sliding relation over another: physical realism should be the deciding factor. A final note is that this issue is not fundamentally specific to the equations solved for ice flow, so while the simulations carried out here use Elmer/Ice to solve the Stokes equations, the same principles should apply in other IDMs that implement some kind of sliding relation.

It might be thought that a special treatment of the grid cell or element containing the grounding line, such that the grounding line position within the cell or element can be represented, would resolve this problem of discretising the friction force feedback. However, the grounding line parameterisations introduced by Gladstone et al. (2010b) (a study featuring numerous different parameterisations implemented in a flowline shelfy-stream model) still show a strong non-linear behaviour correlated to grid cell grounding line advance in the evolution of model state (see Gladstone et al. (2010b) Figures 3, 4 and 6). Gladstone et al. (2010b) also find multiple steady states exist, with one steady state grounding line position per grid cell, although the steady state position is not constrained to lie on a grid point. Thus grounding line parameterisations do not necessarily resolve the problem of a discretised (or highly non-linear on a grid cell scale) friction forcing feedback.

The poor convergence of IDMs regarding grounding line movement emphasizes the important, and hopefully obvious, requirement that all IDM studies featuring grounding line movement demonstrate that sufficiently fine resolution has been used to achieve a converged result. The resolution must be such that the region of multiple steady states, which is known to be a numerical artefact, must collapse toward zero (we suggest as a rule of thumb that it be required to be no larger than one grid cell or element, though this is not itself a formal demonstration of convergence). Reversibility experiments can not in general demonstrate that sufficiently fine resolution has been achieved.

## 5.1 Basal melting

As mentioned above, the forcing feedback involving total basal resistance will be of greatly reduced magnitude in the case of sliding relations in which the basal resistance smoothly transitions to zero approaching the grounding line. Given the currently increasing use of such sliding relations (Gladstone et al., 2017; Brondex et al., 2017; Tsai et al., 2015), it might seem that the impacts on experiment design described here (Section 3.1) will not be widely applicable. However, recent studies have shown that ocean-induced melting at the base of ice shelves can cause convergence problems, both at a sub-grid scale (Seroussi and Morlighem, 2018) and over multiple grid cells or elements (Gladstone et al., 2017). Given that these convergence problems can be partially mitigated by smoothing the basal melt to approach zero approaching the grounding line (in this case approaching from seaward rather than landward), we propose that a forcing feedback, analogous to the friction force feedback, is involved. In this case it is between basal melting and geometry: a retreating grounding line will expose more basal area to melting, increasing thinning in the vicinity of the grounding line and enhancing retreat. Discretisation of this melt forcing feedback also leads to the existence of a region of multiple steady states at coarse resolution (Gladstone et al. (2017), Figure 5), implying that the same restrictions on experiment design will apply.

## 6 Conclusions

The established poor convergence of many marine ice sheet models regarding grounding line movement is characterised by a region of multiple locally stable states. Our results demonstrate that this is not, as has been previously claimed, a neutral equilibrium.

This region of steady states implies that perturbation experiments, such as are often used in model intercomparison projects, can have a hitherto unrecognised dependence on initial conditions, potentially leading to false positives. Thus the size of the region of multiple locally stable steady states may be a more useful metric for assessing convergence of modelled grounding line movement than advance only simulations, retreat only simulations, or reversibility. Specifically, the region should reduce with finer resolution, and should be vanishingly small at published resolutions. If perturbation experiments are used in future tests of convergence of grounding line behaviour, we advocate that the spin-up method must also be prescribed, as a simple requirement for steady state is not in general sufficient to constrain the experiment design.

This poor convergence is not only a result of inherent difficulties in respresenting a spatial step change in basal drag across the grounding line, but is also due to a temporal forcing feedback involving grounding line movement and basal shear stress.

These results apply to marine ice sheet models using a Weertman sliding relation. The same qualitative features occur with sliding relations incorporating a smoother transition in basal drag across the grounding line, such as through a dependence on height above buoyancy, but with a smaller magnitude. A similar convergence issue is raised when imposing basal melt under the floating ice shelf.

*Author contributions.* Rupert Gladstone designed the experiments and led the manuscript writing. Yuwei Xia carried out the simulations and contributed to the manuscript. John Moore contributed to interpretation of results and writing the manuscript.

*Acknowledgements.* This research was supported by Academy of Finland grant number 286587. The authors wish to acknowledge CSC - IT Centre for Science, Finland for computational resources.

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

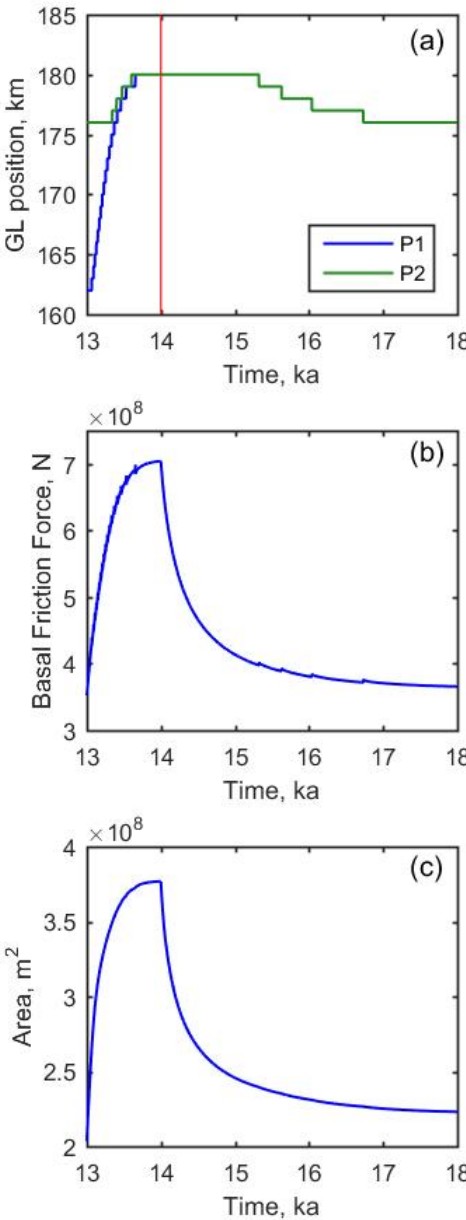

**Figure 3.** Evolution of (a) grounding line position, (b) total basal friction (this is the basal shear stress integrated over the grounded region) and (c) ice area (the flowline equivalent of volume) for perturbation experiments. Both perturbation experiments P1 and P2 are shown in (a), and the difference in initial states is clearly visible. (b) and (c) show only P1. Both P1 and P2 were run for $1\,\mathrm{ka}$ with accumulation rate $a = 2.0\,\mathrm{ma}^{-1}$ followed by $4\,\mathrm{ka}$ with $a = 0.2\,\mathrm{ma}^{-1}$.

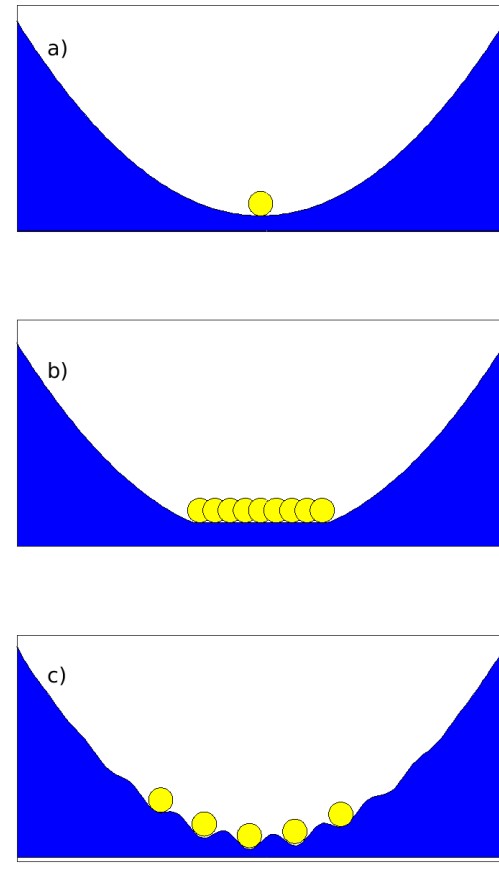

**Figure 4.** A simple idealised system of a ball under gravitation used to demonstrate types of equilibria: (a) one stable equillibrium is present and the ball will always tend to return to this; (b) a region of neutral equilibrium is present (a region of flat surface), and the ball will remain in equilibrium anywhere on this surface; (c) multiple locally stable equilibria exist, and the ball will tend to roll downhill to the nearest.

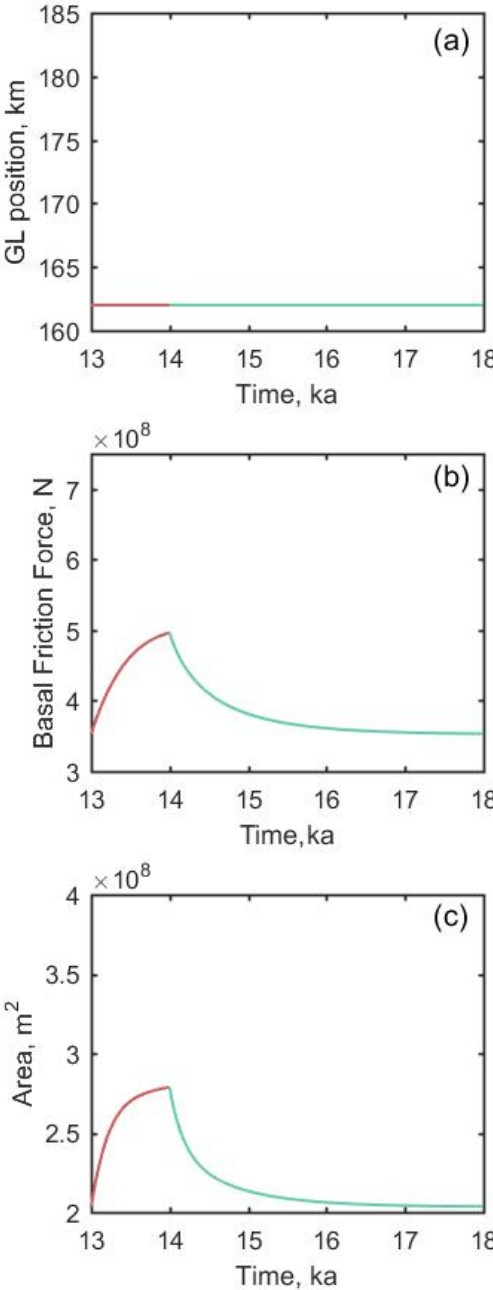

**Figure 5.** Evolution of (a) grounding line position, (b) total basal friction, and (c) ice area (the flowline equivalent of volume) for the small perturbation experiment PS. PS is identical to P1 except that the magnitude of the perturbed forcing for the first 1 ka is $a = 0.71\,\mathrm{ma}^{-1}$ and for the last 4 ka is $a = 0.7\,\mathrm{ma}^{-1}$.

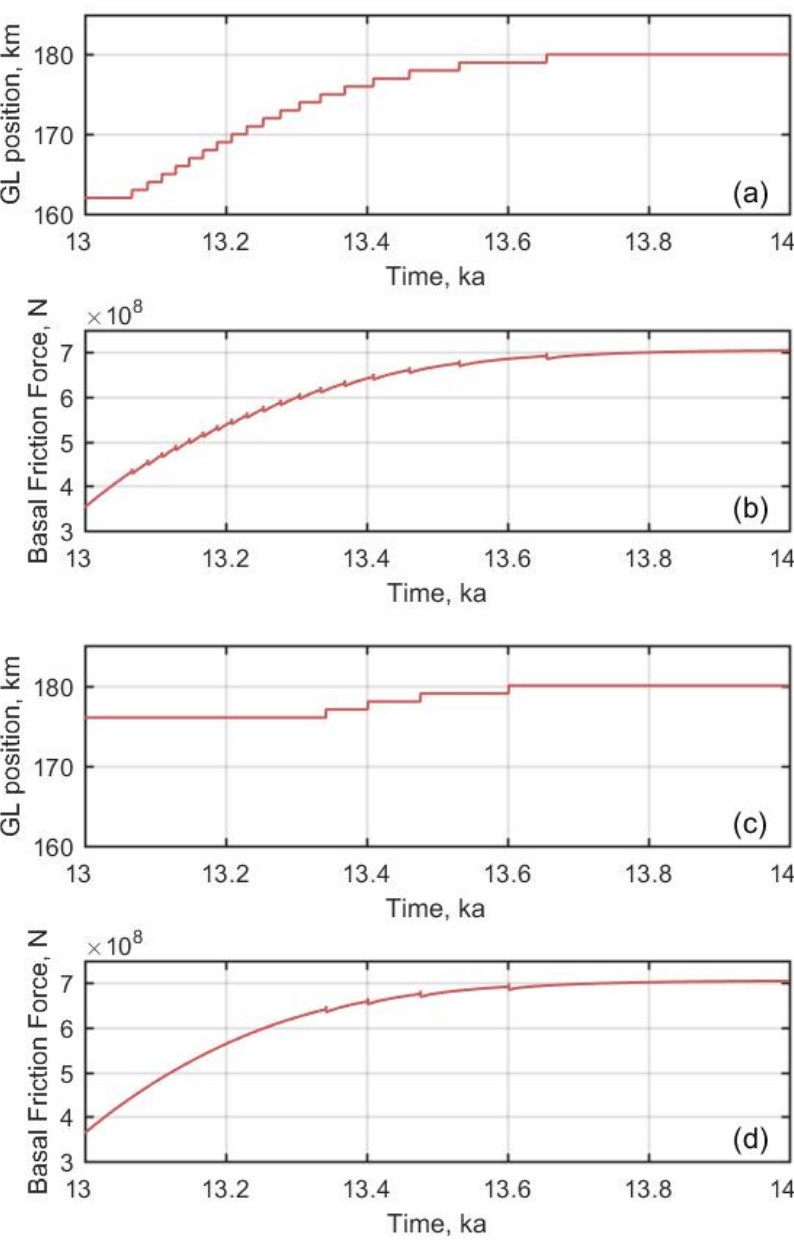

**Figure 6.** A closer look at the advance phase of the perturbation experiments. Evolution of (a) grounding line position for P1, (b) total basal friction for P1, (c) grounding line position for P2, (d) total basal friction for P2.