# Peer review of "Neutral equilibrium and forcing feedbacks in marine ice sheet modelling"

_The Cryosphere, 2018_

## Referee Comment (RC1) · Anonymous Referee #1 · 14 Aug 2018

General comments:

The manuscript discusses the existence of multiple steady state (and poor numerical convergence) in a marine ice sheet grounded on a prograde bedrock slope as a result of the discretisation of the total basal friction, which the authors refer to the friction force feedback. A flowline based on the finite elements method is constructed to solve the full-Stokes equations (in this case, the domain is 2-D with horizontal e vertical axes). Idealized numerical experiments varying the accumulation rate (surface mass balance) are performed to explore the movement of the grounding line in a coarse mesh (the authors purposely defined the model in a coarse resolution, 1 km, to study the behavior of the grounding line). The numerical experiments consist of an advance phase followed by a retreat phase. Two perturbation experiments are also carried out

to investigate further the model reversibility. The grounding line positions as function of simulation time (for all experiments) are shown in graphics. These results are used as arguments in the discussion of the existence of a region multiple steady states, and in the problem of the reversibility. In special, a perturbation experiment (P2) that shows reversibility, even not being in a steady state, is used in the discussion of implications for experiment design. The discussion also counterposes the difference of "neutral equilibrium" and "multiple steady states" (a didactic figure is shown), and a possible reason for the existence of the last one (multiple steady states) and not the first is also listed: the discretisation of the total basal friction (named by the authors as friction force feedback). The result of a small perturbation experiment (PS) where the perturbation force is not sufficient to "move" the grounding line is used as argument of this possible reason.

Overall the manuscript is well written, and the figures are well visible. The experiments are described and constructed to sustain the authors' arguments. The discussion of implications for experiment design to evaluate ice sheet models is relevant. I recommend the publication of the work. Here are some specific comments.

Specific comments:

- The term "false positive" is used in abstract to refer to the case 'that appears to achieve convergence when in fact (...)' is not true. This case is the experiment named P2. The term "false positive" could be written along the text (Sections 2.2, 3.1, 5, 6) such that the reader can link and follow the discussion started in the abstract.

- A suggestion of two additional papers as reference of special treatment of grid cell or element containing the grounding line (page 1, line 24): Seroussi et al. (2014) and Feldmann et al. (2014).

- The term "convergence" is treated as a numerical convergence along all the manuscript. I ask to the authors to insert few words in the beginning of the manuscript (in introduction and maybe in abstract) explaining that the term "convergence" means

(will be used as) "numerical convergence".

- Sections 5 and 6 are not addressed in the end of the Introduction (page 2, lines 9 to 11).

- The lateral drag (an approach to model the buttressing) is parameterised according to channel width. There is no description of how it is done, but it seems that is similar to the reference cited (Gagliardini et al., 2010). An issue that arises is how this lateral drag (buttressing) impact the size of the region of 'multiple steady state'? Did the authors perform any experiments with no lateral drag? For example, experiments P1 and P2 starting since t=0 ka (advance and retreat phases) with no lateral drag?

- It is written (page 2, lines 14 to 15) that the model setup is similar to that original MISMIP. But the bedrock used in the manuscript (Equation 1) is different of the original MISMIP bedrock. The authors could add some words explaining that the bedrock used in the manuscript is inspired or it is a modification of the original benchmark. Is there any reason for that modification?

- What is the length of the domain in the x-direction (maximum x, ice front position)?

- What is the time step used in the experiments? It could be insert in flowline description (Section 2).

- I think it is important to address in the Flowline description (Section 2) that the grounding line position is defined only on the vertices of the elements (I hope I understood correctly the approach that Elmer/Ice does solving the contact problem). Maybe it could be inserted after the description of contact problem (page 3, line 19).

- The experiments (advance/retreat phases) are well written, but a table or a graphic resuming all of them (showing the variation of the external forcing as function of the time, for example) should be also inserted such that the reader will follow the results according. This would enhance the understanding of the results, mainly for P1, P2 and PS (I spent a time following and interpreting the results, the grounding line evolution,

mainly for the perturbation experiments).

- Page 5, line 19. In the phrase: 'The region containing steady state grounding line positions in the current study spans from x = 143 km to x = 176 km'. This interval refers at the end of the retreat phase, right? So, maybe an additional note could be inserted: 'The region containing steady state grounding line positions (at the end of the retreat phase) in the current study spans from x = 143 km to x = 176 km'.

- Page 5, line 21. In the phrase: 'We tested this hypothesis near the seaward end of the region by implementing small increments in a between advance simulations'. Are these increments the simulations with a=1.4 to 2.0 ma-1? If yes, a note could be added, for example: 'we tested this hypothesis near the seaward end of the region by implementing small increments in a between advance simulations (a=1.4 to 2.0 ma-1)'. If no, it is necessary to write more details about what was done.

- Page 5, line 23. Same as above. A note could be added, for example: 'Specifically we obtained a final grounding line position on every node from x = 174 km to x = 180 km for the advance simulations and from x = 174 km to x = 176 km for the retreat simulations (see Figure 2, simulations with a=1.4 to 2.0 ma-1)'.

- It is important to note that between x = 174 km to x = 180 km there are 7 mesh nodes; in x = 174 km to x = 176 km, there are 3 mesh nodes.

- Page 5, line 25. 'even for simulations showing no grounding line movement' -> 'even for simulations showing no grounding line movement (i.e., a=0.5, 0.7, 1.0, 1.4, 1.5, 1.6 ma-1)'

- Page 6, line 5. In the phrase: 'P2 shows full reversibility and P1 does not'. Does this "full reversibility" refer only for the grounding line position or also refer to the ice volume? In 'Figure 3', the variation of the ice volume for P2 is not shown. So, maybe it is also relevant to include (in Figure 3) the variations of P2 in terms of ice volume and basal friction force.

- About the discussion initialized in page 5, line 12 (Section 3.1). The discussion of Schoof (2007) (Section 4.4, page 14, line 105) says that "numerical underresolution may also affect the results of Pattyn et al. (2006) ...". So, what should be the impact of the numerical underresolution on the Pattyn et al. (2006)'s results (in terms of size of region of multiple steady state, reversibility)?

- The phrase (page 7, line 17): "We argue that IDMs exhibit a region containing multiple locally stable equilibria ...". This region exists due to the errors on the numerical modeling, right? I think it should be reinforced in that phrase or in the respective paragraph.

- The fact that some 'IDMs exhibit a region containing multiple locally stable equilibria' makes the "reversibility test" fragile, as the authors well pointed out in Section 5, since the results (reversibility) would depend on the initial condition (the region of multiple locally stable equilibria). However, I think it is important to address in the discussion that the existence of these regions should not be admissible, in the sense that further researcher to improve the numerical schemes used in IDMs should be carried out.

- The phrase (page 8, line 2): "but heavily discretised in the model due to basal friction reaching a peak at the grounding line." Maybe it should be: "but heavily discretised in the model due to the numerical scheme used to solve the contact problem (grounding line "jumps" only on the element nodes)".

- Note that it is expected the basal friction reaches a peak at the grounding line, since it is expected that the basal velocities are higher there (for example, Figure 11 in Schoof (2007)), considering the Weertman model. So, for the flowline-Stokes used in the manuscript, the grounding line represents a "singular point", in the sense that there is an abrupt change in the boundary condition (basal friction) considering the last point grounded (grounding line) and the first floating node (no basal friction). Using another sliding relations, possible this singular point would "vanish", as the authors well written in the paragraph started in line 10, page 8. I recommend the inclusion of these discussions in the manuscript.

[Figure]

- From my point of view, the 'friction force feedback' represents the variation of the boundary condition, which is solution-dependent: depends on the velocity field (in special, the basal velocities) and the position of the last grounded point (grounding line), which in turn depend on the boundary conditions. Then, some possible sources of discretization errors are, in my opinion (not necessarily in this order of weight, and not just summarized to these): a) the boundary condition (the 'friction force feedback' in this case) should be continuous, but it is applied only on the element nodes; b) near (and at) the grounding line, both the velocity field (in special the basal velocity) and the basal friction have high gradients, what could not be well captured if there are few elements in there; c) and, the last grounded node (the grounding line) represents a "singular point", what requires a high mesh resolution in its neighborhood (see, as an example, the Figure 10.9, page 189, in Szabó and Babuska (1991)). So, if the authors agree with my opinion, and if relevant (it is up to the authors), the observations as above could be also added in the discussion part.

- A last question: if the region of multiple steady state is due to the numerical scheme used (so, depends on the IDM), how this region could be used as metric in model evaluation/comparison, as pointed in the conclusion part (page 9, line 2). (If each IDM has its own region of multiple steady state...)

Technical corrections and typos:

- The term 'spin up' is used along all the manuscript. Please, check the correct spelling along all the manuscript ('spinup', 'spin-up' or 'spin up'?): a) page 1, line 12 b) page 5, line 31 c) page 6, line 1, line 3, line 12, line 13, line 17 d) page 9, line 5

- The term 'artifact' is sometimes written as 'artefact'. Please, check the correct spelling along all the manuscript ('artifact' or 'artefact'?): a) page 1, line 3 b) page 5, line 14, line 17 c) page 6, line 19 d) page 8, line 10

- Page 2, line 18. The variable 'W' (channel width) is not used neither defined. Maybe it could be deleted. The channel width should defined in the text.

- In Section 2.1, pg. 5, line 4. 'They are run for 1 ka with a=2.0 ma-1 (...)'. The value of 'a' refers to the forcing perturbation experiments P1 and P2. However, in the legend of Figure 3 (pg. 12), the value of 'a' is 0.2 ma-1. Maybe it is a typo, but I would like to ask to the authors to check if all forcing values (accumulation ratio) are correct.

- Check units and space between number and units in all the text: a) page 2, line 21: -15 C -> -15 oC (please use the default degree symbol of the text editor used) b) page 3, Figure 1: 13ka -> 13 ka; a=0.7ma-1 -> a=0.7 ma-1; a=1.7ma-1 -> a=1.7 ma-1 c) page 3, line 6: 100m -> 100 m d) page 4, Both legends in Figure 2 (accumulation rates legends) e) page 4, Figure 2: 7ka -> 7 ka f) page 4, line 1: 13ka -> 13 ka g) page 5, line 9: 7ka -> 7 ka h) page 5, line 11: 7ka -> 7 ka (maybe here 't=7 ka')

- A note explaining the 'Area' in Figure 5 (c) is the ice volume per unit width should be inserted in the Figure 5 legend (as was written for Figures 2 and 3).

- page 8, line 26: See -> see

- page 7, line 13: Schoof (2007) -> (Schoof, 2007)

References:

Feldmann, J., Albrecht, T., Khroulev, C., Pattyn, F., and Levermann, A.: Resolution-dependent performance of grounding line motion in a shallow model compared with a full-Stokes model according to the MISMIP3d intercomparison, Journal of Glaciology, 60, 353–360, https://doi.org/10.3189/2014JoG13J093, 2014.

Gagliardini, O., Durand, G., Zwinger, T., Hindmarsh, R. C. A., and Le Meur, E.: Coupling of ice-shelf melting and buttressing is a key process in ice-sheets dynamics, Geophysical Research Letters, 37, doi:10.1029/2010GL043334, l14501, 2010.

Pattyn, F., Huyghe, A., De Brabander, S., and De Smedt, B.: Role of transition zones in marine ice sheet dynamics, Journal of Geophysical Research-Earth Surface, 111, doi:10.1029/2005JF000394, 2006.

Seroussi, H., Morlighem, M., Larour, E., Rignot, E., and Khazendar, A.: Hydrostatic grounding line parameterization in ice sheet models, The Cryosphere, 8, 2075–2087, https://doi.org/10.5194/tc-8-2075-2014, 2014.

Schoof, C.: Ice sheet grounding line dynamics: Steady states, stability, and hysteresis, Journal of Geophysical Research-Earth Surface, 112, doi:10.1029/2006JF000664, 2007.

Szabó, B. and Babuška, I.: Finite Element Analysis, John Wiley & Sons, USA, 1991.

---

## Referee Comment (RC2) · Anonymous Referee #2 · 24 Sep 2018

This paper aims to examine grounding-line behavior in advance and retreat scenarios, particularly examining cases in which the author's models show examples of what could be termed neutral stability (or multiple steady-state grounding-line configurations for the same forcing) of grounding line position, while also demonstrating that GL reversibility, in and of itself, is likely not a sufficient test for demonstrating that a model is sufficiently resolved.

The paper is well-written and clear, although would perhaps benefit from a statement of the goals of the experiments at the beginning. The approach taken is well-described (I think I could re-run these experiments on my own if I wanted to), and the figures are for the most part clear and well-documented (the figure illustrating stability is a useful one). It is a useful addition to the literature, and I support publication after a few fairly

minor points are addressed.

I think the biggest thing missing from this paper is much, if any, discussion of mesh resolution. It's not controversial to state that an insufficiently-resolved ice sheet model will exhibit artifacts in its grounding-line response (even alluded to that in the discussion). It would be very helpful to present some sort of mesh convergence result to demonstrate the regime being operating in for this paper. Resolution is mentioned at the beginning (implying operation in an under-resolved regime), but then don't do anything to place the experiments in context in this sense. Without that sort of discussion, it's tempting to label the results here as "odd things that happen when a grounding-line problem is under-resolved"), and attribute the multiple steady-states to hysteresis due to under-resolution. It would be very useful if you picked a few of the initial cases (say a= 0.2, 0.7, and 1.7) and show (a) the convergence of the GL and area at steady-state with mesh resolution, and then (b) how the experiments behave in fully- and under-resolved regimes. Otherwise, you essentially seem to be making the point that GL reversibility is not a sufficient test by itself to demonstrate that a model is sufficiently-resolved (which is an important point – that the only reliable way to assess whether one is sufficiently-resolved is via a convergence study along the lines of Cornford et al (2016) – but it's not a point that's being made explicitly in this paper).

The role of mass balance also isn't mentioned in the results – You appear to have chosen a test case in which the additional mass flux onto the grounded ice due to the increased surface area for an advanced grounding line is exactly balanced by the increased flux through the GL due to the increased ice thickness at the GL (hence the apparent multiple stability points). A useful test would be to try this experiment again with (for example) a different bed slope, which would in principle change that relationship.

**1 Specific comments:**

1. page 1, line 9, 11, 16, etc: The word "convergence" has a particular meaning in numerical modeling describing how a model behaves as the mesh spacing, timestep, etc are refined (or possibly the tendency of, say, a solver, to reduce its residual to a prescribed tolerance). In at least some places, you appear to use "convergence" when you likely mean "steady state". I'd suggest a careful check on all of the uses of the word "convergence" to ensure that it's being used consistently. Otherwise, there is a tendency for confusion when a single word has multiple meanings and connotations. I'd even suggest the use of "convergence with resolution", etc...

2. page 2, line 8: I'd suggest also citing Seroussi and Morlighem (2018) on discretizing melt forcing near grounding lines here.

3. page 2, line 9: I think the choice of flowline modeling and Weertman sliding law are unfortunate here – flowline because it's perhaps overly simplistic given the current understanding of the effect of buttressing and other effects that are not present in a flowline model; I would have suggested using either a MISMIP3D or MISMIP+ configuration as a testbed. Weertman is unfortunate because as the authors point out, it produces much more of a forcing discontinuity at the grounding line, which is likely amplifying the effects described in this work; something like the Tsai Coulomb-limited sliding law would have been a useful counterpoint. That said, none of these specifically discount the conclusions drawn in this paper, but instead leave important questions unexamined.

4. page 2: (problem description) – how long is your domain in the x-direction?

5. Figure 1:

    (a) Is there really no vertical shear in the velocity field? That's surprising, but is the impression I get from the vertically-constant coloring of the velocity field.

    (b) The use of the intensity-based colormap doesn't work well with the two-profile plot as designed, since the semi-transparent colors can't be distinguished from different speeds for the second profile. I'd suggest switching to a colormap which isn't intensity-based if you want to present the second profile as a lighter-shaded version of the primary colormap. Another option would be to simply show only one representative velocity magnitude plot, but overlay the outlines (in black) of multiple profiles, which would also allow for more than one alternative profile.

6. Figure 2:

    (a) Did the 0.2 m/a run ever actually achieve steady-state? It's not obvious from the area plot.

    (b) It appears that all of the cases for which the advance-phase a is greater than 1.6 m/a all collapse onto the 1.6 profile. Is that the case? I'd say this is really odd behavior; do you have any idea why? Are the GL's all getting stuck on the same cell boundary? It seems like the behavior is very different above and below that threshold.

7. Section 3 (discussion of multiple steady states) – it might make sense to move this section to before what is now section 2.2; if you did that, the initial experiment (section 2.1) would be followed by the discussion of the initial experiment, and then the follow-on perturbation experiments would have a context when they're introduced. It would also lessen the number of times a reader has to page back and forth between experiment description, results, and discussion.

8. page 5, line 10: A useful test for steady-state would be to compute the time derivative of the area and plot that.

9. page 5, line 10: "grouding"-> "grounding"

10. page 7, line 2: suggest changing "a ball on a hill" to "a ball perched on the summit of a hill"

11. page 7, line 2: "after begin" -> "after being"?

12. page 7, line 6: suggest changing "A large perturbation" to "A large-enough perturbation"

13. page 8, line 25: This suggests something isn't quite right, since one would expect the friction to vary smoothly throughout advance and retreat if the subgrid-scale friction discretization is done correctly. If that's the case, then one wouldn't expect to see mesh-related artifacts of advance and retreat in the friction field, would you?

14. Figure 3: Why isn't experiment P2 shown in 3(b) and 3(c)?

15. Figure 5: What happens if you allow the system to reach steady-state after the perturbation is applied (rather than discontinuing the forcing after 1000 years? Does the GL finally advance, in that case?

**2  References:**

1. Cornford, S., Martin, D., Lee, V., Payne, A.,  Ng, E. (2016). "Adaptive mesh refinement versus subgrid friction interpolation in simulations of Antarctic ice dynamics". *Annals of Glaciology*, **57**(73), 1-9. doi:10.1017/aog.2016.13

2. Seroussi, H. and Morlighem, M., (2018), "Representation of basal melting at the grounding line in ice flow models", *The Cryosphere Discussions*, **2018**, 1–16, doi:10.5194/tc-2018-117

---

## Author Response (AR1)

**Author responses for "Neutral equilibrium and forcing feedbacks in marine ice sheet modelling"**

Rupert Gladstone, Yuwei Xia and John Moore

October 21, 2018

**1  Author notes**

The authors would like to thank the reviewers for their constructive comments and suggestions. We have copied the text from the reviewers below. Authors responses are given in blue. *Page and line numbers of additions/modifications in the revised manuscript are given in blue italics.* We provide in addition both the modified manuscript and a document highlighting the changes between our original and revised manuscripts.

**2  Response to RC1**

**2.1  General comments**

The manuscript discusses the existence of multiple steady state (and poor numerical convergence) in a marine ice sheet grounded on a prograde bedrock slope as a result of the discretisation of the total basal friction, which the authors refer to the friction force feedback. A flowline based on the finite elements method is constructed to solve the full-Stokes equations (in this case, the domain is 2-D with horizontal e vertical axes). Idealized numerical experiments varying the accumulation rate (surface mass balance) are performed to explore the movement of the grounding line in a coarse mesh (the authors purposely defined the model in a coarse resolution, 1 km, to study the behavior of the grounding line). The numerical experiments consist of an advance phase followed by a retreat phase. Two perturbation experiments are also carried out to investigate further the model reversibility. The grounding line positions as function of simulation time (for all experiments) are shown in graphics. These results are used as arguments in the discussion of the existence of a region multiple steady states, and in the problem of the reversibility. In special, a perturbation experiment (P2) that shows reversibility, even not being in a steady state, is used in the discussion of implications for experiment design. The discussion also counterposes the difference of "neutral equilibrium" and "multiple steady states" (a didactic figure is shown), and a possible reason for the existence of the last one (multiple steady states) and not the first is also listed: the discretisation of the total basal friction (named by the authors as friction force feedback). The result of a small perturbation experiment (PS) where the perturbation force is not sufficient to "move" the grounding line is used as argument of this possible reason.

Overall the manuscript is well written, and the figures are well visible. The experiments are described and constructed to sustain the authors arguments. The discussion of implications for experiment design to evaluate ice sheet models is relevant. I recommend the publication of the work.

Thanks to the reviewer for positive review, and for the constructive comments. These have improved the clarity of the manuscript.

**2.2  Specific comments**

- The term "false positive" is used in abstract to refer to the case that appears to achieve convergence when in fact (...) is not true. This case is the experiment named P2. The term "false positive" could be

written allong the text (Sections 2.2, 3.1, 5, 6) such that the reader can link and follow the discussion started in the abstract.

*We have now made use of this term in sections 3.1 and 6 (note that section 2.2 is purely descriptive where as the term "false positive" is an interpretation, and that section 5 is mainly about the friction feedback).*

*Page 6, line 31; page 11, line 2*

- A suggestion of two additional papers as reference of special treatment of grid cell or element containing the grounding line (page 1, line 24): Seroussi et al. (2014) and Feldmann et al. (2014).

*yes, these are also relevant, and we've added them now.*

*Page 2, lines 11-12; page 10, lines 20-21; page 2, line 2*

- The term "convergence" is treated as a numerical convergence along all the manuscript. I ask to the authors to insert few words in the beginning of the manuscript (in introduction and maybe in abstract) explaining that the term "convergence" means (will be used as) "numerical convergence".

*We have clarified our use of the term convergence where it first occurs in the introduction*

*Page 1, lines 20-22*

- Sections 5 and 6 are not addressed in the end of the Introduction (page 2, lines 9 to 11).

*We've added a line at the end of the inctroduction to refer to section 5. Section 6 (conclusions) needs no introducing.*

*Page 2, lines 17-18.*

- The lateral drag (an approach to model the buttressing) is parameterised according to channel width. There is no description of how it is done, but it seems that is similar to the reference cited (Gagliardini et al., 2010). An issue that arises is how this lateral drag (buttressing) impact the size of the region of multiple steady state? Did the authors perform any experiments with no lateral drag? For example, experiments P1 and P2 starting since t=0 ka (advance and retreat phases) with no lateral drag?

*This is a mistake in our description. In fact we did some initial simulations in which there was lateral drag, but all the simulations shown have zero lateral drag. We've removed the mention of lateral drag.*

*Informally I can say that the presence of lateral drag can reduce the relative impact of the step change in resistive forces at the grounding line, reducing the relative importance of the friction feedback, and reducing the size of the region of steady states. However, the lateral drag needs to be very high in order to make a difference when a Weertman sliding relation is used. This argument is already made in the Gladstone et al. 2012 Annals paper.*

- It is written (page 2, lines 14 to 15) that the model setup is similar to that original MISMIP. But the bedrock used in the manuscript (Equation 1) is different of the original MISMIP bedrock. The authors could add some words explaining that the bedrock used in the manuscript is inspired or it is a modification of the original benchmark. Is there any reason for that modification?

*We don't feel that the values of parameters chosen relative to previous experiments are of much importance. The relevant thing here is that the inputs we use demonstrate the features (irreversibility, region of steady states, advance and retreat) of relevance. We've extended the sentence about MISMIP to summarise the similarities, but we don't feel a discussion of the differences adds value to the paper.*

*Page 2, lines 22-23*

- What is the length of the domain in the x-direction (maximum x, ice front position)?

*We should have stated this. It is 600km. We've added this information just below the bedrock equation.*

*Page 3, line 3*

- What is the time step used in the experiments? It could be insert in flowline description (Section 2).

*We should have stated this. It is 0.2 years. We've added this information in section 2.*

*Page 3, lines 13-14*

- I think it is important to address in the Flowline description (Section 2) that the grounding line position is defined only on the vertices of the elements (I hope I understood correctly the approach that Elmer/Ice does solving the contact problem). Maybe it could be inserted after the description of contact problem (page 3, line 19).

*Yes, this is correct. We've clarified it and added a line about represnting friction at the grounding line.*

*Page 2, lines 27-30*

- The experiments (advance/retreat phases) are well written, but a table or a graphic resuming all of them (showing the variation of the external forcing as function of the time, for example) should be also inserted such that the reader will follow the results according. This would enhance the understanding of the results, mainly for P1, P2 and PS (I spent a time following and interpreting the results, the grounding line evolution, mainly for the perturbation experiments).

It is a bit awkward to make the table because the advance retreat experiments are almost identical, and having a row each for this is just a waste of space. We've implemented a name convention for these to enable making the table, and we've added the table. We've modified the advance retreat section to introduce the new naming.

*Page 3, Table 1; Page 3, lines 22-24; Page 4, lines 3-4*

Note that we also made a minor mistake describing the small perturbation experiment, implying that it had the same retreat forcing as P1 and P2. We've corrected this.

*Page 4, lines 9-11*

- Page 5, line 19. In the phrase: The region containing steady state grounding line positions in the current study spans from x = 143 km to x = 176 km. This interval refers at the end of the retreat phase, right? So, maybe an additional note could be inserted: The region containing steady state grounding line positions (at the end of the retreat phase) in the current study spans from x = 143 km to x = 176 km.

We've added this as suggested.

*Page 6, line 7*

- Page 5, line 21. In the phrase: We tested this hypothesis near the seaward end of the region by implementing small increments in a between advance simulations. Are these increments the simulations with a=1.4 to 2.0 ma-1? If yes, a note could be added, for example: we tested this hypothesis near the seaward end of the region by implementing small increments in a between advance simulations (a=1.4 to 2.0 ma-1). If no, it is necessary to write more details about what was done.

Yes. We've clarified this using the new experiment naming.

*Page 6, line 10*

- Page 5, line 23. Same as above. A note could be added, for example: Specifically we obtained a final grounding line position on every node from x = 174 km to x = 180 km for the advance simulations and from x = 174 km to x = 176 km for the retreat simulations (see Figure 2, simulations with a=1.4 to 2.0 ma-1).

We've added a reference to the figure.

*Page 6, line 12*

- It is important to note that between x = 174 km to x = 180 km there are 7 mesh nodes; in x = 174 km to x = 176 km, there are 3 mesh nodes.

We're not convinced this is important to point out. In any case it is self-evident, given that the mesh resolution is 1km. We have not altered the text.

- Page 5, line 25. even for simulations showing no grounding line movement -¿ even for simulations showing no grounding line movement (i.e., a=0.5, 0.7, 1.0, 1.4, 1.5, 1.6 ma-1)

Again, we've followed this suggestion using the new experiment naming.

*Page 6, line 14*

- Page 6, line 5. In the phrase: P2 shows full reversibility and P1 does not. Does this "full reversibility" refer only for the grounding line position or also refer to the ice volume? In Figure 3, the variation of the ice volume for P2 is not shown. So, maybe it is also relevant to include (in Figure 3) the variations of P2 in terms of ice volume and basal friction force.

The ice volume and friction are mostly shown to support later arguments. Showing both P1 and P2 for these plots is not very useful because the differences are at the far left couple of mm of the plot, and are obscured by the rapid rates of change. We've clarified that reversibility does refer to all aspects of the model state.

*Page 6, lines 29-30*

- About the discussion initialized in page 5, line 12 (Section 3.1). The discussion of Schoof (2007) (Section 4.4, page 14, line 105) says that "numerical underresolution may also affect the results of Pattyn et al. (2006) ...". So, what should be the impact of the numerical underresolution on the Pattyn et al. (2006)s results (in terms of size of region of multiple steady state, reversibility)?

Previous work by Gladstone et al (notably JGR 2010, TC 2010, Annals 2012) has demonstrated that the region of steady states in general decreases in size with finer resolution. This is not specific to the Pattyn 2006 paper.

The current study demonstrates that reversibility is not a robust test of whether sufficiently fine resolution has been achieved. Again, this is not specific to the Pattyn 2006 study. We have added a line to make this clearer in the previous paragraph.

*Page 6, lines 31-34*

The important implications for the Pattyn 2006 study are that their experiment design did not reliably support their result about transition zone length, and we believe we have demonstrated this clearly.

- The phrase (page 7, line 17): "We argue that IDMs exhibit a region containing multiple locally stable equilibria ...". This region exists due to the errors on the numerical modeling, right? I think it should be reinforced in that phrase or in the respective paragraph.

Yes, it is a numerical artefact. This is an important point which appears to have confused the other reviewer. We've modified the first and last sentences of the relevant paragraph to make this clear.

*Page 8, lines 25 and 28*

- The fact that some IDMs exhibit a region containing multiple locally stable equilibria makes the "reversibility test" fragile, as the authors well pointed out in Section 5, since the results (reversibility) would depend on the initial condition (the region of multiple locally stable equilibria). However, I think it is important to address in the discussion that the existence of these regions should not be admissible, in the sense that further researcher to improve the numerical schemes used in IDMs should be carried out.

Certainly any numerical artefacts should ideally be removed from, or at least quantified in, any published studies using IDMs to make statements about the real world. Hopefully this is obvious to all readers, but we've added a paragraph at the end of the discussion just to make this absolutely clear.

*Page 10, lines 9-14*

- The phrase (page 8, line 2): "but heavily discretised in the model due to basal friction reaching a peak at the grounding line." Maybe it should be: "but heavily discretised in the model due to the numerical scheme used to solve the contact problem (grounding line "jumps" only on the element nodes)".

The suggested change in wording implies that the problem is specific to the numerical scheme used in this particular model. But this is not the case, the problem is common to nearly all models, and the various parameterisations used are in general only partial solutions. For this reason we prefer the original wording.

- Note that it is expected the basal friction reaches a peak at the grounding line, since it is expected that the basal velocities are higher there (for example, Figure 11 in Schoof (2007)), considering the Weertman model. So, for the flowline-Stokes used in the manuscript, the grounding line represents a "singular point", in the sense that there is an abrupt change in the boundary condition (basal friction) considering the last point grounded (grounding line) and the first floating node (no basal friction). Using another sliding relations, possible this singular point would "vanish", as the authors well written in the paragraph started in line 10, page 8. I recommend the inclusion of these discussions in the manuscript.

This is a valid point, but is not the main focus of the current manuscript. These issues are discussed in more detail in more relevant recent papers (Gladstone et al 2017 in TC and Brondex et al 2017, J. Glac.).

- From my point of view, the friction force feedback represents the variation of the boundary condition, which is solution-dependent: depends on the velocity field (in special, the basal velocities) and the position of the last grounded point (grounding line), which in turn depend on the boundary conditions. Then, some possible sources of discretization errors are, in my opinion (not necessarily in this order of weight, and not just summarized to these): a) the boundary condition (the friction force feedback in this case) should be continuous, but it is applied only on the element nodes; b) near (and at) the grounding line, both the velocity field (in special the basal velocity) and the basal friction have high gradients, what could not be well captured

if there are few elements in there; c) and, the last grounded node (the grounding line) represents a "singular point", what requires a high mesh resolution in its neighborhood (see, as an example, the Figure 10.9, page 189, in Szab and Babuska (1991)). So, if the authors agree with my opinion, and if relevant (it is up to the authors), the observations as above could be also added in the discussion part.

These are valid points, but they all essentially relate to resolution and or grounding line parameterisations, which are the focus of other more relevant studies. In particular Gagliardini 2016 TC considered approaches to handling the singularity in the Elmer/Ice model, but plenty of other studies over the last decade look at this issue and how different approaches have been taken in different models. There are different angles one could take when looking at this issue, and noone yet has a really good solution. We prefer not to modify the text in response to this comment as we want to keep the focus of this paper on the issue of the numerical artefacts and neutral equilibrium in relation to experiment design.

- A last question: if the region of multiple steady state is due to the numerical scheme used (so, depends on the IDM), how this region could be used as metric in model evaluation/comparison, as pointed in the conclusion part (page 9, line 2). (If each IDM has its own region of multiple steady state...)

Well, it is a numerical artefact, and so it should reduce with increasingly finer resolution. Any model can demonstrate this without recourse to another model. And of course it would be possible to compare the size of the region across models at a given resolution. We've added a sentence to the conclusions, though we don't feel it necessary to point out the latter point, as this may be misleading: if model A is "better" than model B at resolution XXX, but model B is typically run at resolutions several orders of magnitude finer than XXX, then such a comparison is not meaningful. Also, such a direct comparison is not informative about the rate of convergence.

*Page 11, lines 4-5*

**2.3   Technical corrections and typos**

- The term spin up is used along all the manuscript. Please, check the correct spelling along all the manuscript (spinup, spin-up or spin up?):
a) page 1, line 12
b) page 5, line 31
c) page 6, line 1, line 3, line 12, line 13, line 17
d) page 9, line 5

We now use only "spin-up".

- The term artifact is sometimes written as artefact. Please, check the correct spelling along all the manuscript (artifact or artefact?):
a) page 1, line 3
b) page 5, line 14, line 17
c) page 6, line 19
d) page 8, line 10

Both spellings are ok. We have now used "artefact" everywhere.

- Page 2, line 18. The variable W (channel width) is not used neither defined. Maybe it could be deleted. The channel width should defined in the text.

Removed.

- In Section 2.1, pg. 5, line 4. They are run for 1 ka with a=2.0 ma-1 (...). The value of a refers to the forcing perturbation experiments P1 and P2. However, in the legend of Figure 3 (pg. 12), the value of a is 0.2 ma-1. Maybe it is a typo, but I would like to ask to the authors to check if all forcing values (accumulation ratio) are correct.

We've corrected the figure caption. Thanks!

*Page 15, Figure 3 caption*

- Check units and space between number and units in all the text:
a) page 2, line 21: -15 C -¿ -15 oC (please use the default degree symbol of the text editor used)

b) page 3, Figure 1: 13ka -¿ 13 ka; a=0.7ma-1 -¿ a=0.7 ma-1; a=1.7ma-1 -¿ a=1.7 ma-1
c) page 3, line 6: 100m -¿ 100 m
d) page 4, Both legends in Figure 2 (accumulation rates legends)
e) page 4, Figure 2: 7ka -¿ 7 ka
f) page 4, line 1: 13ka -¿ 13 ka
g) page 5, line 9: 7ka -¿ 7 ka
h) page 5, line 11: 7ka -¿ 7 ka (maybe here t=7 ka)

We use Latex. We've now tried to make our units and spacing consistent, and will also follow the recommendations of the Copernicus typesetters if the manuscript is accepted.

- A note explaining the Area in Figure 5 (c) is the ice volume per unit width should be inserted in the Figure 5 legend (as was written for Figures 2 and 3).

Good point. We've implemented this change.
*Page 17, Figure 5 caption*
- page 8, line 26: See -¿ see
Changed as requested.
*Page 19, line 5*
- page 7, line 13: Schoof (2007) -¿ (Schoof, 2007)
Changed as requested.
*Page 8, line 24*

**3 Response to RC2**

**3.1 General comments**

This paper aims to examine grounding-line behavior in advance and retreat scenarios, particularly examining cases in which the authors models show examples of what could be termed neutral stability (or multiple steady-state grounding-line configurations for the same forcing) of grounding line position, while also demonstrating that GL reversibility, in and of itself, is likely not a sufficient test for demonstrating that a model is sufficiently resolved.

The paper is well-written and clear, although would perhaps benefit from a statement of the goals of the experiments at the beginning. The approach taken is well-described (I think I could re-run these experiments on my own if I wanted to), and the figures are for the most part clear and well-documented (the figure illustrating stability is a useful one). It is a useful addition to the literature, and I support publication after a few fairly minor points are addressed.

Thanks to the reviewer for the constructive comments. We've expanded the final paragraph of the introduction to give a clearer context for our experiments.
*Page 2, lines 14-18*

I think the biggest thing missing from this paper is much, if any, discussion of mesh resolution. Its not controversial to state that an insufficiently-resolved ice sheet model will exhibit artifacts in its grounding-line response (even alluded to that in the discussion). It would be very helpful to present some sort of mesh convergence result to demonstrate the regime being operating in for this paper. Resolution is mentioned at the beginning (implying operation in an under-resolved regime), but then dont do anything to place the experiments in context in this sense. Without that sort of discussion, its tempting to label the results here as "odd things that happen when a grounding-line problem is under-resolved"), and attribute the multiple steady-states to hysteresis due to underresolution. It would be very useful if you picked a few of the initial cases (say a= 0.2, 0.7, and 1.7) and show (a) the convergence of the GL and area at steady-state with mesh resolution, and then (b) how the experiments behave in fully- and under-resolved regimes. Otherwise, you essentially seem to be making the point that GL reversibility is not a sufficient test by itself to demonstrate that a model is sufficiently-resolved (which is an important point  that the only reliable way to assess whether

one is sufficiently resolved is via a convergence study along the lines of Cornford et al (2016) but its not a point thats being made explicitly in this paper).

*It is not true to say that discussion of mesh resolution is missing from this paper. Whenever we mention convergence, which is discussed quite extensively, we mean the convergence of model outputs with respect to increasingly finer resolution. We have added clarification in the introduction that this is what we mean by convergence.*

*Page 1, lines 20-22*

*A misunderstanding appears to have occurred here. It is not merely tempting but actually quite correct to label the results here as "odd things that happen when a grounding-line problem is under-resolved", and to attribute the multiple steady-states to hysteresis due to underresolution. Exploring numerical artefacts is a primary aim of this paper, because of their relevance to understanding both the nature of the grounding line problem and implications for designing computer experiments. We think the enhanced final paragraph of the introduction will help to make this clear. We have also enhanced the section on neutral equilibrium, explicitly stating twice that the region of steady states is an artefact.*

*Page 8, lines 25 and 28*

*Actually the Cornford 2016 convergence test is not necessarily robust because of the way it is initialised. Using present day geometry and inversion for basal resistance does not provide any guarantee of where the initial grounding line position may lie with respect to a potential region of multiple stable grounding line positions. We have added a couple of paragraphs on the implications of our results for simulations initialised through inversion in section 3.1.1.*

*Page 7, lines 15-30.*

The role of mass balance also isnt mentioned in the results You appear to have chosen a test case in which the additional mass flux onto the grounded ice due to the increased surface area for an advanced grounding line is exactly balanced by the increased flux through the GL due to the increased ice thickness at the GL (hence the apparent multiple stability points). A useful test would be to try this experiment again with (for example) a different bed slope, which would in principle change that relationship.

*The chance of the balance described by the reviewer actually occurring is vanishingly small. Of course increased grounded area must lead to increased steady state thickness at the grounding line, but in practice this could only lead to a neutral equilibrium in the underlying system with careful engineering (of, for example, spatially varying basal resistance). In fact Schoof (2007) showed that the relationship between grounding line ice thickness and cross-grounding line flux is strongly non-linear. Multiple steady state grounding line positions have been reported before with different models and with different (but also linear) bed slopes. For example see other Gladstone et al papers (in particular JGR 2010, TC 2017). The multiple steady states are due to numerical artefacts, as we aim to describe in the sections about the friction feedback.*

**3.2   Specific comments**

1. page 1, line 9, 11, 16, etc: The word "convergence" has a particular meaning in numerical modeling describing how a model behaves as the mesh spacing, timestep, etc are refined (or possibly the tendency of, say, a solver, to reduce its residual to a prescribed tolerance). In at least some places, you appear to use "convergence" when you likely mean "steady state". Id suggest a careful check on all of the uses of the word "convergence" to ensure that its being used consistently. Otherwise, there is a tendency for confusion when a single word has multiple meanings and connotations. Id even suggest the use of "convergence with resolution", etc...

*We always mean "convergence with resolution" and have clarified this in the introduction. We have double checked all instances of the word and these are all correct.*

*Page 1, lines 20-22*

2. page 2, line 8: Id suggest also citing Seroussi and Morlighem (2018) on discretizing melt forcing near grounding lines here.

We have added a line about the Seroussi paper, which is a useful addition to the literature. We've also extended the discussion to consider the implications of a basal melt forcing feedback.

*Page 2, lines 11-12; Page 10, lines 15-27*

3. page 2, line 9: I think the choice of flowline modeling and Weertman sliding law are unfortunate here flowline because its perhaps overly simplistic given the current understanding of the effect of buttressing and other effects that are not present in a flowline model; I would have suggested using either a MISMIP3D or MISMIP+ configuration as a testbed. Weertman is unfortunate because as the authors point out, it produces much more of a forcing discontinuity at the grounding line, which is likely amplifying the effects described in this work; something like the Tsai Coulomb-limited sliding law would have been a useful counterpoint. That said, none of these specifically discount the conclusions drawn in this paper, but instead leave important questions unexamined.

In order to demonstrate a specific feature of a system, the greatest clarity and attribution can be achieved by demonstrating the simplest configuration in which said feature is manifest. This, plus computational efficiency, motivates our choice for a flowline domain.

Weertman sliding was chosen for 2 reasons: it is commonly used and it gives strong numerical artefacts, which are, after all, what we wish to focus on. The results shown here do apply, with a lesser magnitude, to other sliding relations, and this is discussed in the paper.

4. page 2: (problem description) how long is your domain in the x-direction?

600km, we've added this information now.

*Page 3, line 3*

5. Figure 1:

(a) Is there really no vertical shear in the velocity field? Thats surprising, but is the impression I get from the vertically-constant coloring of the velocity field.

There is vertical shear, but the velocity variations are dominated by increase along the flow, which is much greater than the vertical shear. Bear in mind also that the plot is vertically exaggerated. We have now mentioned this in the Figure caption.

*Page 4, Figure 1 caption*

(b) The use of the intensity-based colormap doesnt work well with the two profile plot as designed, since the semi-transparent colors cant be distinguished from different speeds for the second profile. Id suggest switching to a colormap which isnt intensity-based if you want to present the second profile as a lighter-shaded version of the primary colormap. Another option would be to simply show only one representative velocity magnitude plot, but overlay the outlines (in black) of multiple profiles, which would also allow for more than one alternative profile.

The aim of the profile plot is to give the reader an intuitive view of the model setup. We don't need to show more than the two profiles used in the perturbation experiments. The plot conveys the convex nature of the grounded region and it conveys that the ice velocity increases down stream, and it conveys the magnitude of difference between initial states for hte perturbation experiment. There is no further information that needs to be conveyed by this plot. It is not important to show the details of the velocity profiles separately. We have not changed this plot.

6. Figure 2:

(a) Did the 0.2 m/a run ever actually achieve steady-state? Its not obvious from the area plot.

This simulation had an unchanging grounding line position for the last 6000 years. Also, we added a line to report rates of change of area at the end of the simulations, to demonstrate that steady state has been approached.

*Page 5, lines 2-4*

(b) It appears that all of the cases for which the advance-phase a is greater than 1.6 m/a all collapse onto the 1.6 profile. Is that the case? Id say this is really odd behavior; do you have any idea why? Are the GLs all getting stuck on the same cell boundary? It seems like the behavior is very different above and below that threshold.

Yes, all GLs on the same node at 176km. This is the upper end of the region of steady state grounding line positions, as discussed in section 3. We believe this is due to discretisation of a link between model state and forcing, which is discussed in sections 4 and 5. We've added a couple of lines to the end of section 3 to clarify the link between this behaviour and our explanation.

*Page 6, lines 16-19*

7. Section 3 (discussion of multiple steady states) it might make sense to move this section to before what is now section 2.2; if you did that, the initial experiment (section 2.1) would be followed by the discussion of the initial experiment, and then the follow-on perturbation experiments would have a context when theyre introduced. It would also lessen the number of times a reader has to page back and forth between experiment description, results, and discussion.

This is a valid point, but there are arguments both ways. An early draft of the paper was laid out as suggested, but co-authors found it counter-intuitive. The current layout may require some referencing back and forth, but on the plus side it is clear where to find the information needed, on account of having one section to describe the model and experiments. We prefer to keep the current structure.

8. page 5, line 10: A useful test for steady-state would be to compute the time derivative of the area and plot that.

We have now calculated the area change rate and find it to be negligible for our advance and retreat simulations. We report this in Section 3.

*Page 5, lines 2-4*

9. page 5, line 10: "grouding"-¿ "grounding"

Fixed, thanks.

10. page 7, line 2: suggest changing "a ball on a hill" to "a ball perched on the summit of a hill"

Changed as requested.

*Page 8, line 13*

11. page 7, line 2: "after begin" -¿ "after being"?

Fixed, thanks.

*Page 8, line 14*

12. page 7, line 6: suggest changing "A large perturbation" to "A large-enough perturbation"

Changed to "A sufficiently large perturbation"

*Page 8, line 17*

13. page 8, line 25: This suggests something isnt quite right, since one would expect the friction to vary smoothly throughout advance and retreat if the subgrid-scale friction discretization is done correctly. If thats the case, then one wouldnt expect to see mesh-related artifacts of advance and retreat in the friction field, would you?

Indeed, the problem here is that noone has yet come up with the perfect grounding ine parameterisation, and perhaps such a thing is simply not possible. People have certainly come up with "correct" implementations of an interpolation based on the floatation condition, but it doesn't fully capture the forcing feedback, perhaps partly because the ice geometry at sub grid scale is not captured. If you can come up with a grounding line parameterisation in which mesh artefacts are absent then you'll be famous (as Steve Price once said to me)!

14. Figure 3: Why isnt experiment P2 shown in 3(b) and 3(c)?

It is barely distinguishable from P1, which we felt was more confusing than including it. In fact the aim of subplots (b) and (c) is to start discussion on the forcing feedback. Subplot (a) is already sufficient to demonstrate the weakness of perturbation experiments as a demonstration of convergence.

15. Figure 5: What happens if you allow the system to reach steady-state after the perturbation is applied (rather than discontinuing the forcing after 1000 years? Does the GL finally advance, in that case?

It is actually fairly close to steady state after 1ka. But it is not essential to reach steady state to demonstrate that this is not a neutral equilibrium, which is the main purpose of PS.

**Neutral equilibrium and forcing feedbacks in marine ice sheet modelling**

Rupert Gladstone[1], Yuwei Xia[2], and John Moore[1,2]

[revised manuscript text omitted]